# Automated Detection of Conifer Seedlings in Drone Imagery Using Convolutional Neural Networks

**Michael Fromm** [1,*]**, Matthias Schubert** [1]**, Guillermo Castilla** [2] **, Julia Linke** [3]
**and Greg McDermid** [3]

1    Institute for Informatic, Ludwig-Maximilians-Universität München, Oettingenstraße 67,
D-80333 Munich, Germany; schubert@dbs.ifi.lmu.de

2    Northern Forestry Centre, 5320 122 Street Northwest, Edmonton, AB T6H 3S5, Canada;
guillermo.castilla@canada.ca

3    Department of Geography, University of Calgary, Calgary, AB T2N 1N4, Canada;
julia.linke@ucalgary.ca (J.L.); mcdermid@ucalgary.ca (G.M.)

*    Correspondence: fromm@dbs.ifi.lmu.de

**Abstract:** Monitoring tree regeneration in forest areas disturbed by resource extraction is a requirement for sustainably managing the boreal forest of Alberta, Canada. Small remotely piloted aircraft systems (sRPAS, a.k.a. drones) have the potential to decrease the cost of field surveys drastically, but produce large quantities of data that will require specialized processing techniques. In this study, we explored the possibility of using convolutional neural networks (CNNs) on this data for automatically detecting conifer seedlings along recovering seismic lines: a common legacy footprint from oil and gas exploration. We assessed three different CNN architectures, of which faster region-CNN (R-CNN) performed best (mean average precision 81%). Furthermore, we evaluated the effects of training-set size, season, seedling size, and spatial resolution on the detection performance. Our results indicate that drone imagery analyzed by artificial intelligence can be used to detect conifer seedlings in regenerating sites with high accuracy, which increases with the size in pixels of the seedlings. By using a pre-trained network, the size of the training dataset can be reduced to a couple hundred seedlings without any significant loss of accuracy. Furthermore, we show that combining data from different seasons yields the best results. The proposed method is a first step towards automated monitoring of forest restoration/regeneration.

**Keywords:** convolutional neural networks; forest restoration; regeneration surveys; seedling detection; UAVs; sRPAS

## 1. Introduction

### 1.1. Introduction

Restoration and reforestation of forest areas subject to anthropogenic disturbance must be accompanied by effective monitoring of the survival and establishment of tree seedlings in those areas. Traditionally, monitoring programs have relied on costly field surveys that involve identifying and counting seedlings on the ground. However, recent advances in platform technology, sensor systems, and photogrammetry may allow drastic time and cost savings for monitoring forest regeneration. Imagery can be collected by digital cameras mounted on small remotely piloted aircraft systems (sRPAS, aka unmanned aerial vehicles (UAVs) or drones [1]). Afterwards the collected imagery can be processed using structure-from-motion (SfM [2]) techniques and/or modern object-detection methods [3] based on convolutional neural networks (CNNs) to assess the amount and status of regeneration.

In this paper, we apply CNN-based object-detection methods to drone imagery for automatically detecting conifer seedlings along recovering seismic lines: a common legacy footprint from oil and gas exploration [4].

*1.2. Remote Sensing of Forest Regeneration: Related Work*

Early work on the assessment of forest regeneration using remote sensing was based on manual photointerpretation of large-scale aerial photographs acquired from helicopters. Using 1:500 color-infrared photography, Hall et al. [5] managed to detect just 44% of small seedlings (crown diameter < 30 cm) in harvest blocks in Saskatchewan, Canada, in the early 1990s. Twenty-five years later, Goodbody et al. [6] classified color imagery of 2.4 cm ground sampling distance (GSD) acquired from a rotary wing sRPAS over harvest blocks replanted 5 to 15 years earlier in British Columbia, Canada, and obtained user accuracies for plot-level coniferous cover between 35% and 97% using image segmentation and random forest classification. However, no count of individual seedlings was attempted. Puliti et al. [7] estimated stem density in 580 small (50 $m^2$) circular plots in young conifer stands (mean height 2.5 m, mean density 5479 stems/ha) in Stange, Norway, using random forest with predictors derived from 3 cm GSD drone imagery and obtained a RMSE for stand density of 21.8%. While far more accurate than visual estimates from foresters or even estimates from airborne laser scanning (ALS) data, their method cannot provide information on the location of individual seedlings. Feduck et al. [8] used a simple workflow to detect individual small coniferous seedlings in harvest blocks in Alberta, Canada, using 3 mm GSD color imagery from a sRPAS, and achieved a 75% detection rate using image segmentation and a classification and regression tree (CART). While their method can provide the approximate location of the detected seedlings, it relies on seasonal leaf-off conditions when the conifer seedlings are spectrally distinct from their surroundings.

We could not find any peer-review reference on the use of CNNs for conifer seedling detection (there is, however, a growing number of papers on the use of CNN for the detection of seedlings and weeds in agricultural settings (e.g., [9–11]). Notwithstanding, New Zealand's Forestry Research Institute, a.k.a. Scion, has recently announced that it has developed a deep learning algorithm that uses UAV imagery to identify radiata pine seedlings [12], but the actual research has not been published yet. For a general review of the use of CNNs in remote sensing, see [13] and [14].

While the detection of mature individual trees with CNNs have been successfully attempted for plantations of oil palm trees [15], and for urban trees [16], the detection of individual specimens of a particular tree species in a natural setting is a far more complex task. Perhaps the first example of this in the peer-reviewed literature is Morales et al. [17], who used drone imagery of 1.4 to 2.5 cm GSD and a Google's Deeplab v3+ network to detect aguaje palm trees (*Mauritia flexuosa*) in the Amazon, and obtained very accurate results similar to hand-drawn maps.

*1.3. Contributions of Our Approach*

In this paper, we show for the first time in the literature that it is indeed possible to operationally detect conifer seedlings in millimetric (GSD < 1 cm) drone imagery using CNN-based object-detection methods. Unlike other features of seismic lines like coarse woody debris, detecting conifer seedling poses several challenges. In particular, conifer seedlings strongly vary in size, color, and shape. In order to provide guidelines for forest restoration assessment using CNNs, we examined the impacts on the detection performance of ground sampling distance, the amount of annotated seedlings for training, and the phenological conditions (leaf-on vs. leaf-off) of the site when imaging occurred. The output of the detection can be summarized in a map with the location and size of individual seedlings, which can be used to compute insightful local and global statistics on regeneration and identify areas requiring further restoration. Given that the system can be applied to images from the same area over several years, it is also possible to follow the development of the detected seedlings through time, and thus monitor their survival and growth.

The paper is structured as follows: We first describe the study area, the collected imagery and the process of annotating training data for the object detector. Afterwards, we survey the components and techniques of the evaluated object detection methods. Finally, we present various experimental settings and their observed results and derive some conclusions on the best combination of techniques and on the influence of characteristics such as phenological conditions, the amount of training data, and pixel and seedling size.

## 2. Materials and Methods

### 2.1. Study Area

The study area for this research is located in the boreal forest of northeastern Alberta, near the town of Conklin (Figure 1). The location is in the Central Mixedwood natural subregion [18], which is characterized by a mix of upland and lowland forests over gently undulating terrain. Upland forests are comprised mainly of jack pine (*Pinus banksiana*), black spruce (*Picea mariana*), and trembling aspen (*Populus tremuloides*) stands, while wetlands support black spruce, tamarack (*Larix laricina*), and various non-tree species [18]. The area also contains a variety of human disturbances related to forestry and petroleum development, including roads, well sites, forest-harvest areas, and seismic lines (petroleum-exploration corridors). Seismic lines are particularly common, being found in densities approaching 10 km/km$^2$ in the region. These are of particular interest, given their role in the decline of boreal woodland caribou (*Rangifer tarandus caribou*), a species listed under the federal Species at Risk Act [19]. Seismic lines fragment caribou habitat, increasing access for other ungulate species and their predators [20], which in turn increases predation rates for caribou. As a result, fostering return to forest cover on disturbance features such as seismic lines is seen as a key element of caribou conservation, and the focus of seismic-restoration efforts [21].

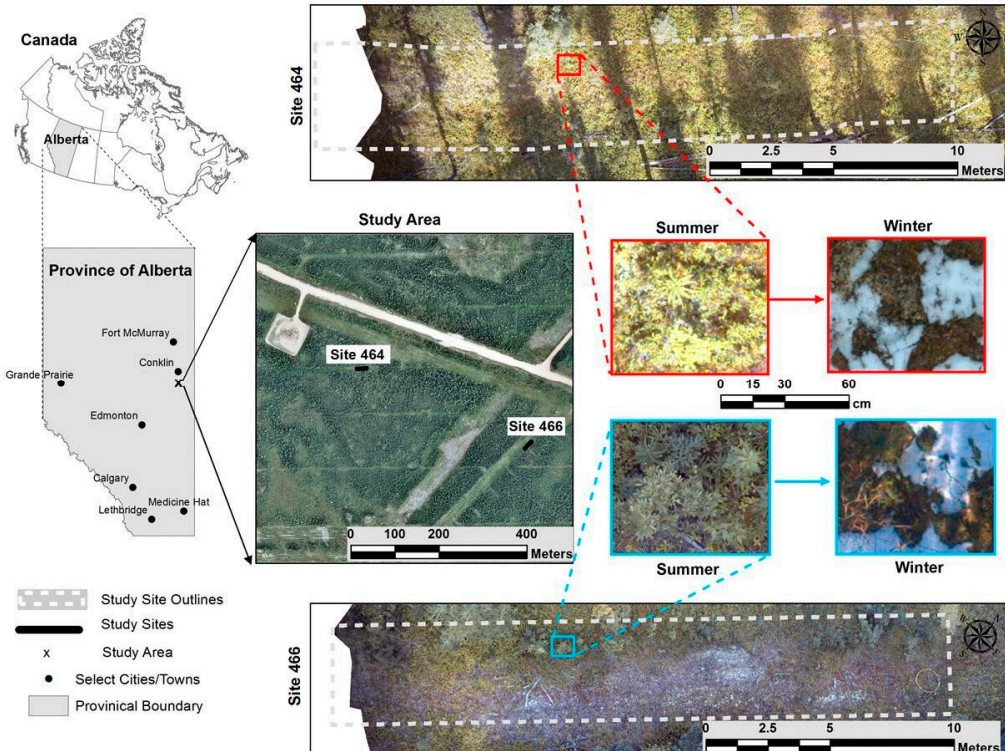

**Figure 1.** Experiments took place at two sites in the boreal forest south of Conklin Alberta, which we refer to as site 464 and site 466. Both are short (~25 m) segments of seismic lines (petroleum-exploration corridors) containing regenerating seedlings. Imagery over the sites was obtained in both leaf-on ('summer') and leaf-off ('winter') conditions.

Our flights took place within experimental sites established by the Boreal Ecosystem Recovery and Assessment (BERA) project (Figure 1). The two sites selected for this work are sites 464 and 466: ~25 m long sections of seismic lines containing regenerating seedlings, which we are familiar with due to ongoing work within BERA.

## 2.2. Image Data

The images used in this study were acquired by a DJI Mavic Pro flying at about 5 m above ground level on two different dates: 3 August 2017 (hereafter 'summer'), and 20 October 2017 (hereafter 'winter'). At this low altitude, the FC220 camera (4.73 mm focal length) inbuilt in the drone, which was pointing straight down, yielded a ground sampling distance (GSD) of 0.3 cm at nadir. We note that flying above the canopy would have resulted in imagery of a few centimeters GSD, which we initially thought may be too coarse for our application, hence our choice of flying inside the line. Sky was mostly clear at both dates and sun elevation was 50° in summer and 20° in winter.

## 2.3. Image Preprocessing

We cropped all captured images into multiple smaller non-overlapping tiles, as these require less memory on the graphics processing unit (GPU) that was used for training the model. An additional advantage of this method is that the process of annotating conifer seedlings was easier on smaller tiles because it simplifies the process of finding all seedlings over a limited sample area. We chose a tile size of 512 pixels by 512 pixels, which offered a good compromise between the average size of a seedling in the image and the size of the full image. The average size of the bounding box around a seedling in our imagery was about 100 pixels by 100 pixels. Thus, the number of seedlings split between adjacent tiles was limited. By cropping the images into tiles, we also lost parts of the original image on the right and lower sides of the image, which did not contain enough pixels to build a complete tile. However, the images taken by the drone usually overlap; thus, the missing information on the border of the images was negligible.

## 2.4. Image Annotation

After image preprocessing, we used the graphical image annotation tool LabelImg [22] to manually draw bounding boxes and assign labels around 200 seedlings, mostly black spruce, which was the most prevalent seedling species in the lines we surveyed. Based on this limited set of bounding boxes, we trained a first object-detection model as described in Section 2.5 and applied it to the remaining images to create a set of candidate boxes. Although the detected boxes were preliminary and included many false positives, they helped to reduce the amount of manual labor involved in the annotation process. The automatically generated candidate boxes were loaded into LabelImg with the default labeling, then manually corrected by a human analyst. This way, the analyst only had to adjust the location of the correctly detected bounding boxes, delete the false positives, and add a limited number of boxes around undetected seedlings. In this fashion, a total amount of 3940 conifer seedlings were annotated in the 9415 tiles we had (Table 1).

Since images in the dataset often showed the same seedling from multiple camera locations, we assembled the training and test set manually to avoid having the same seedling appear in both sets. We decided to use all images from site 466, all the summer images from site 464 and half the winter images from site 464 to build the training set (1863 image tiles). The test set consisted of half the winter images from site 464 and 661 image tiles. All images in the train- and test datasets contain at least one seedling.

**Table 1.** Overview of the different training- and test sets, where all images include at least one seedling.

| Experiment | Train/Test | Site 464 | | | | Site 466 | | | | Total | |
| | | Winter (leaf-off) | | Summer (leaf-on) | | Winter (leaf-off) | | Summer (leaf-on) | | | |
| | | Images | Tiles | Images | Tiles | Images | Tiles | Images | Tiles | Images | Tiles |
|---|---|---|---|---|---|---|---|---|---|---|---|
| Summer | Train | | | 37 | 387 | | | 14 | 132 | 51 | 519 |
| | Test | | | | | | | 21 | 215 | 21 | 215 |
| Winter | Train | | | | | 59 | 670 | | | 59 | 670 |
| | Test | 86 | 896 | | | | | | | 86 | 896 |
| Summer -> Winter | Train | | | 37 | 387 | | | 35 | 347 | 72 | 734 |
| | Test | 116 | 1177 | | | 82 | 894 | | | 198 | 2071 |
| Winter -> Summer | Train | 116 | 1177 | | | 82 | 894 | | | 198 | 2071 |
| | Test | | | 37 | 387 | | | 35 | 347 | 72 | 734 |
| Combination | Train | 32 | 235 | 37 | 387 | 82 | 894 | 35 | 347 | 186 | 1863 |
| | Test | 54 | 661 | | | | | | | 54 | 661 |

## *2.5. Automated Seedling Detection Architectures*

### 2.5.1. Object Detection Architectures

In this work, we examined the use of three state-of-the-art object detectors: faster-region convolutional neural network (faster R-CNN) [23], region-based fully convolutional network (R-FCN) [24] and single shot multibox detector (SSD) [25] to detect conifer seedlings on the dataset described above. Each detector extracts image features using a convolutional neural network (CNN) to generate an abstract image representation called a feature map. Based on this feature map, the detectors examine multiple sub images potentially containing an object of interest, predict object classes, and draw minimal bounding boxes around the detected objects. Since the object detector is independent of the feature map, multiple combinations of object detector and CNN are possible. In general, the architectures can be trained to simultaneously detect various types of objects, e.g., cars, pedestrians, and traffic signs. However, in our application, the only object class was conifer seedling. As a result, we did not use this multi-combination functionality. All of the examined architectures were trained using stochastic gradient descent to optimize the weights determining the output of the nodes in the neural network. We used the Adam [26] optimizer to adjust learning rates during the optimization to improve convergence. Since all weights within the network are trained on images and annotated bounded boxes, the architectures were trained end-to-end. In other words, there were no intermediate states during the detection process.

Faster-RCNN and R-FCN are two-stage object detectors that employ the following stages:

(1) The model proposes a set of candidate regions of interest by a select search [27] or a regional proposal network [23].

(2) A succeeding neural network is used on the candidate regions of interest to decide the class and the most likely bounding box around the object.

Two-stage detectors are usually very accurate and allow for a good detection of overlapping objects. However, they also require large computational resources, longer optimization times, and are less efficient when applied to new images.

SSD is a one-stage detector that examines a set of default bounding box positions on a set of feature maps corresponding to a variety of image resolutions to compensate scaling differences. One-stage detectors often provide less accurate results, but also require less complex architectures, less resources, and are more efficient in their application to new images than two-stage detectors. This is of particular interest if images have to be analyzed on-site or in real-time within embedded devices, like the camera of a drone.

As mentioned in the previous section, the named object detection architectures can be combined with various CNNs for feature-map generation. Most CNNs differ in the number of layers, intermediate features, and the resulting inference and training speed (see Table 2). In Section 3.2 we compared results

of seedling detection with four different CNNs: Inception v2 [28], ResNet-50 [29], ResNet-101 [29], and Inception ResNet v2 [30].

**Table 2.** List of available pre-trained object detectors in the TensorFlow Object Detection Package. Inference speed, number of parameters of the models in million, the number of layers in the network and the mean-average-precision (MAP) on the standard object-detection benchmark Microsoft Common Objects in Context (MS COCO) are reported. MS COCO numbers are based on the Tensorflow Object Detection Benchmarks.

| Convolutional Net | Speed (ms) | Parameters | Layers | MS COCO MAP |
|---|---|---|---|---|
| Inception v2 | 58 | 10 M | 42 | 28 |
| ResNet-50 | 89 | 20 M | 50 | 30 |
| ResNet-101 | 106 | 42 M | 101 | 32 |
| Inception ResNet v2 | 620 | 54 M | 467 | 37 |

Networks based on Inception modules [28] provide convolution blocks stacked on each other. Inside the blocks, the method uses $1 \times 1$ convolutions to reduce the number of parameters of the CNN.

Residual networks [29] offer the freedom to skip certain layers during training. In deeper architectures, a CNN with residual connections often performs better than networks with Inception modules. This is due to the degradation problem with deeper networks (for more information see [29]). The Inception ResNet v2 [30] introduces residual connections into the Inception v2 Architecture.

Summarizing, the examined object detection architectures varied with respect to the architecture used to detect objects and the basic CNN for feature map generation. Additionally, we examined whether pre-training and data augmentation had a positive effect on the detection rate.

### 2.5.2. Pretrained Feature Maps

Since most datasets for object detection comprise a limited number of training images, it is common practice to employ CNNs that have been already trained for image classification. This makes sense because the first layers of a CNN represent low-level image features, such as edges and textures that can be found on different types of images. Thus, pre-training the CNN on a large image dataset like the MS COCO [31] usually results in more stable feature maps.

After pre-training, the weights of the trained CNN can be used to initialize the weights of the lower levels of an object detector. This method is often referred to as transfer learning because the weights of the lower levels are transferred from a network pre-trained on a different task and data set.

### 2.5.3. Data Augmentation

Another useful technique in settings with a limited number of training samples is data augmentation. The idea behind this technique is to apply various image transformations to the input images to increase the size of the training set. Examples for these kinds of transformations are rotations, flips, and cropping. Data augmentation also helps prevent the neural network from overfitting to the training set.

For the SSD object detector, we used random horizontal flips and random crops as data-augmentation options. Since the SSD object detector has difficulties with small objects [25], the use of random crops works as a zoom mechanism and increases the performance of the object detector. For faster R-CNN and R-FCN, we used random horizontal flips and random 90 degrees rotations as data augmentation options.

### 2.5.4. Seasonal Influence

Another interesting question is whether seedling detection is more accurate on images from leaf-on (summer) or leaf-off (winter) period (see Figure 1 summer and winter insets). To assess this, we annotated the images of both study sites in summer and winter. The manual annotation of summer

images turned out to be more difficult compared to winter images, because of surrounding green vegetation. On the other hand, winter images often showed larger areas covered with snow, which might conceal small seedlings. To assess seasonal influence, we trained and evaluated the object detectors using either only summer images, only winter images, or both. This allowed us to examine the question whether training on images from various different contexts is beneficial compared to specializing on imagery from a particular time of the year. Furthermore, we wanted to examine which imagery is more suitable for generating accurate results.

### 2.5.5. Emulated Flying Altitude and Ground Sampling Distance

Our data set consisted of images taken at 5 m flying altitude, resulting in a ground sampling distance of 0.3 cm. To emulate the effect of flying at higher altitudes yielding larger ground sampling distances, we artificially blurred and downscaled the images in several stages and examined the detection rate based on the changed ground sampling distance. To emulate different flying altitudes, images were blurred using a Gaussian filter with a standard deviation of σ = 1/s and then averaged over s × s windows where s is the desired scale ratio. This process approximates images shot at an increased flying altitude but does not consider the alteration of perspective. We conducted experiments with s ∈ {5, 9, 21}, which resulted in images simulated from 25 m, 45 m, and 105m altitude, or 1.5 cm, 2.7 cm, and 6.3 cm ground sampling distance respectively (Figure 2).

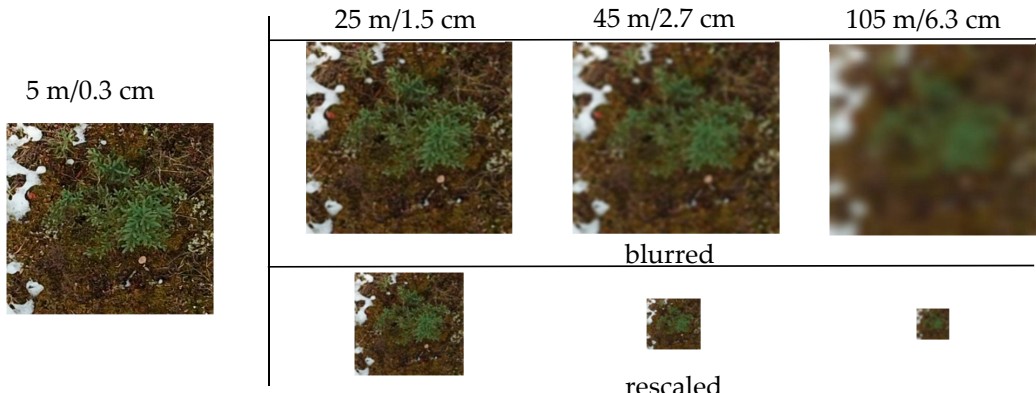

**Figure 2.** Comparison of images at several emulated flying altitudes. The upper row shows the result of the employed blurring process whereas the lower row additionally shows the size the image would have on an image taken at the corresponding altitude.

### 2.5.6. Training Set Size

To determine the number of annotation boxes required to train a reliable object detector (i.e., a detector with mean average precision (MAP) > 0.5), we prepared several training sets providing an increasing number of tiles and annotation boxes. In addition, we wanted to examine the effect of pre-training when smaller training sets are used.

### 2.5.7. Seedling Size

To get a better insight on the sources of detection errors, we further examined the impact of the size of the seedling on the MAP. For this, we split the evaluation set into three subsets according to the area of the bounding boxes. The first subset comprised 197 seedlings deemed small. A seedling was considered small if its surrounding bounding box contained less than 7000 pixels, at a ground sampling distance of 0.3 cm per pixel. This included all seedlings having a bounding box smaller than 30 cm × 30 cm in the physical world. The medium subset consisted of 128 seedlings with bounding boxes containing more than 7000 pixels and less than 35,000 pixels. This corresponded to boxes of up to 66 cm × 66 cm. The large dataset consisted of seedlings bigger than 66 cm × 66 cm (*n* = 36). The final

subset represented 36 seedlings with large bounding boxes, which comprised all boxes having more than 35,000 pixels.

*2.6. Evaluation Metrics for Automated Seedling Detection*

The result of applying an object detector as described above to an image is a set of bounding boxes. For each box, the detector provides a confidence value indicating the likelihood that the object in the box is a conifer seedling. To measure the quality of detection, we had to compare these predicted boxes with the manually drawn annotation boxes describing the true position of the seedlings. Thus, we needed to first decide whether the predicted box had a high-enough confidence value to deem it a 'detected seedling', and then check whether it sufficiently overlapped with a ground-truth box to deem it a 'true detection'. The most common evaluation measure for object detection considering both aspects is the mean average precision (MAP).

To measure the overlap between output and reference, we used the intersection over union (IoU), i.e., the ratio between the union and the intersection of predicted boxes and ground truth boxes. Given two bounding boxes, A (the ground truth) and B (the predicted bounding box), the IoU is defined as:

$$IoU = A \cap B / A \cup B \tag{1}$$

where the intersection operation is defined as the number of pixels that rectangles A and B have in common, and the union operation is defined as the combined number of pixels from A and B.

To compute MAP, we only considered those predicted boxes having an IoU of at least a certain threshold $\tau_{IoU}$ ($\tau_{IoU} = 0.5$ in most cases). Thus, if a box does not overlap with any ground-truth box with an IoU of more than $\tau_{IoU}$, it is considered as a false positive (FP). In terms of the confidence values for each box, we considered any box having less confidence than a confidence threshold $\tau_c$ as FP as well. Correspondingly, any box with $IuO > \tau_{IoU}$ and a confidence larger than $\tau_c$ was considered a true positive (TP). Finally, we considered false negatives as those ground-truth boxes, which were not found by the detector. This can either occur if there was no predicted box with an $IoU > \tau_{IoU}$ or if the confidence value of sufficiently overlapping boxes were too small, i.e., less than $\tau_c$. Given these cases, we computed the precision (P) as P = TP/TP + FP. In other words, P measures how many of the detected seedlings are true seedlings. Correspondingly, the recall (R) is the proportion of true seedlings the detector could find (R = TP/TP + FN).

Precision and recall strongly depend on the confidence threshold $\tau_c$. A smaller value $\tau_c$ usually increases the recall by reducing the amount of false negatives. On the other hand, it may also increase the number of false positives and thus reduce the precision. Hence, there is a trade-off between both measures, which can be visualized by a precision–recall curve. This curve plots the largest achievable precision for a given recall value. The curve was computed by step-wise increasing $\tau_c$ and plotting the precision if a further increase would lower the recall.

Now, to have a measure summarizing the precision–recall curve into a single number, we computed the average precision (AP) over all recall values. In general, the AP is a metric for a single class and in cases we need to take the mean average precision (MAP) for all classes the detector can distinguish. However, since we only considered a single class in our setting, AP and MAP were identical.

We evaluated several combinations of CNN and object detectors and examined the use of pre-training and data augmentation on these architectures. The experiments were run on a workstation with an Intel I7 processor 128Gb main memory and two NVidia GTX 1080Ti GPUs. We used the TensorFlow Object Detection Package (https://github.com/tensorflow/models/tree/master/research/object_detection) to implement all employed neural networks.

## 3. Results

Our best-performing architecture was a faster R-CNN object detector in combination with a pre-trained ResNet-101 CNN to generate the feature maps. This architecture achieved a MAP of

0.81 when using the complete data set and pre-training on the MS COCO dataset [31]. In the next subsections we provide a detailed comparison of various combinations and present the impact of modifications of the data set on the detection performance.

### 3.1. Influence of the CNN Used to Generate the Feature Map

We conducted multiple tests for studying the influence that the CNN generating the feature map has on the MAP of the object detector. For these experiments, we selected faster R-CNN as an object detector due to its widespread use and highly accurate object detection reported for various applications. We compared Inception v2 [28], ResNet-50 [29], ResNet-101 [29], and Inception ResNet v2 [30] to ascertain whether layer depth and model complexity have an advantageous effect on the detector performance (Table 3).

**Table 3.** Comparison of multiple faster R-CNN backbones. The evaluation metric is MAP@0.5IoU.

| Object Detector | Layers | Parameters | COCO MAP | Seedling MAP |
|:---:|:---:|:---:|:---:|:---:|
| Inception v2 | 42 | 10 M | 0.28 | 0.66 |
| ResNet-50 | 50 | 20 M | 0.30 | 0.66 |
| ResNet-101 | 101 | 42 M | 0.32 | 0.81 |
| Inception ResNet v2 | 467 | 54 M | 0.72 | 0.71 |

Regarding CNNs with few layers, Inception v2 achieved comparable performance to ResNet-50, although it had just half the parameters (NB. For comparison, the second last column is the CNN accuracy on the commonly used object detection dataset COCO). Regarding deeper architectures, ResNet-101 performed better than Inception ResNet v2 although it had fewer parameters. As a result, we can state that the CNN used for generating the feature map has a major impact on the performance as ResNet-101 outperformed the other architectures by at least 0.1.

### 3.2. The Effect of Pre-Training

We evaluated the impact of pre-training on the examined object detectors SSD, R-FCN, and faster R-CNN. For our experiments, we compared untrained CNNs with CNNs pre-trained on the COCO dataset [31]. Note that even though Inception ResNet v2 displayed a better classification accuracy than ResNet-101 on the COCO dataset (Table 3), ResNet-101 still proved to generate better feature maps for object detection. Therefore, we used ResNet-101 to assess the effect of pre-training on object detection on R-FCN and faster R-CNN. For SSD, we used pre-trained weights from Inception Net v2 architecture. We employed a more lightweight CNN for SSD because its simpler one-stage architecture would lose its efficiency advantages when combined with a rather deep and complex CNN for feature generation. Table 4 shows the effects of pre-training on the MAP of the three object detection architectures. Faster R-CNN and R-FCN benefit from the pre-trained CNN feature maps by increasing the MAP score by 0.10 and 0.03, respectively. However, the use of a pre-trained CNN for SSD decreased the performance of the detector by 0.07. This might be the result of the shallower architecture of SSD and Inception Net v2. Given the reduced number of layers and weights, it can be expected that all layers of the SSD are less general and more problem-specific. Thus, the lower levels of SSD most likely need to learn more problem-specific features due to the level of abstraction within the network. Thus, pre-training on general images rather decreased the performance of the single-stage detector.

**Table 4.** Comparison of different object detectors on the conifer seedling dataset. The evaluation used MAP@0.5IoU as metric.

| Object Detector | Without Pre-Training | With Pre-Training |
|:---:|:---:|:---:|
| SSD | 0.65 | 0.58 |
| R-FCN | 0.68 | 0.71 |
| Faster R-CNN | 0.71 | 0.81 |

*3.3. Data Augmentation*

In the following, we present the results of our experiments with respect to data augmentation (Table 5).

**Table 5.** Comparison of different object detectors with and without data augmentation. The metric is MAP@0.5IoU.

| Object Detector | No Augmentation | Data Augmentation |
|:---:|:---:|:---:|
| SSD | 0.58 | 0.69 |
| R-FCN | 0.71 | 0.76 |
| Faster R-CNN | 0.81 | 0.80 |

For the SSD, data augmentation resulted in a large increase of 0.11 in MAP. In an additional analysis, we saw that the performance increase was largely based on the performance on smaller seedlings (+0.27). Thus, the cropping augmentation worked to compensate the weakness on very small objects.

R-FCN also benefited from the random rotations and horizontal flips with an increase of 0.05. Interestingly, Faster R-CNN did not profit from data augmentation in our experiments, which might be due to the generality of the region of interest proposal layers, which already copes quite well with varying object positions.

*3.4. Seasonal Influence*

The performance of the different experiments all showed a lower performance of object-detection methods when trained and evaluated on images from a single season than when the detectors were trained on data from both seasons (Table 6). This result stresses that robustness against different image contexts decreased with narrow training data. Thus, we could conclude that it was beneficial to include images from a variety of contexts to achieve robust results on new images. In addition, the experiments on the summer images showed lower performance decreases compared to those trained on the full data set. This is counterintuitive to the experience that the manual annotation was more difficult on the summer images. However, due to the limited size of the training set, the observed differences might depend on sample bias.

**Table 6.** Comparison of different object detectors between summer and winter seasons. The metric is MAP@0.5IoU.

| Object Detector | Summer | Winter | Both |
|:---:|:---:|:---:|:---:|
| SSD | 0.45 | 0.41 | 0.65 |
| R-FCN | 0.69 | 0.59 | 0.71 |
| Faster R-CNN | 0.71 | 0.65 | 0.81 |

*3.5. Emulated Flying Altitude and Ground Sampling Distance*

In order to understand how the altitude and, therefore, the ground-sampling distance, influences the object detection performance, we conducted experiments on down-sampled images. We distinguished between two settings:

Training and testing down-sampled images (Table 7).

**Table 7.** Comparison of object detectors between different resolutions in the first setting, where we trained and evaluated the down-sampled images. The metric is MAP@0.5IoU.

| Object Detector | 5 m/0.3 cm | 25 m/1.5 cm | 45 m/2.7 cm | 105 m/6.3 cm |
|---|---|---|---|---|
| SSD | 0.58 | 0.56 | 0.50 | 0.45 |
| R-FCN | 0.71 | 0.66 | 0.67 | 0.57 |
| Faster R-CNN | 0.81 | 0.75 | 0.72 | 0.66 |

Training on the 5 m images and only testing the down-sampled images (Table 8).

**Table 8.** Comparison of object detectors between different resolutions in the second setting, where we only evaluated the down-sampled images. The metric is MAP@0.5IoU.

| Object Detector | 5 m/0.3 cm | 25 m/1.5 cm | 45 m/2.7 cm | 105 m/6.3 cm |
|---|---|---|---|---|
| SSD | 0.58 | 0.56 | 0.17 | 0.00 |
| R-FCN | 0.71 | 0.72 | 0.16 | 0.00 |
| Faster R-CNN | 0.81 | 0.78 | 0.22 | 0.02 |

The object detector performance decreased gradually with increasing ground-sampling distance (i.e., with decreasing spatial resolution) across all object detectors. In the first setting, corresponding to training and testing on the down-sampled images, the results indicate that object detectors were able to detect conifer seedlings reliably even on images having a simulated altitude of 45 m (i.e., some 20 m above the forest canopy), which corresponds to a ground sampling distance of 2.7 cm. However, when applying the detector trained on high resolution images to the coarser data emulating higher altitudes, the performance quickly decreased for higher simulated altitudes. The effects can be explained by the fact that patterns of seedling on lower resolution look more and more different. Thus, an object detector cannot learn to detect these unknown patterns due to their absence in the training data.

*3.6. Dataset Size*

In general, the more complex two-stage detectors F-RCN and faster-R-CNN showed promising results even for a limited data set of only 500 samples. SSD performed the worst on reduced training datasets, although increasing the number of samples showed a steady performance increase. Given the trend for each detector, it seems reasonable to expect that providing more samples will further increase the MAP scores of all detectors.

R-FCN and faster R-CNN benefited the most (see Table 9) from using pre-trained architectures: their performance on 200 seedlings was comparable to their performance without pre-training on the whole dataset (3940 seedlings). SSD also benefited slightly from pre-training in most cases. The comparison on the whole dataset for SSD shows a decrease in performance, which might depend on the shallower architecture as described in Section 3.3. In general, pre-training was helpful to improve the detection performance in particular for the settings with small training sets.

**Table 9.** Comparison of different object detectors based on dataset size. The metric is MAP@0.5IoU.

| Object Detector | Pre-Trained | 200 | 500 | 1000 | 2000 | 3940 |
|---|---|---|---|---|---|---|
| SSD | () | 0.15 | 0.24 | 0.36 | 0.48 | 0.65 |
| R-FCN | () | 0.39 | 0.49 | 0.58 | 0.61 | 0.68 |
| Faster R-CNN | () | 0.52 | 0.59 | 0.62 | 0.65 | 0.71 |
| SSD | (×) | 0.16 (+1%) | 0.30 (+6%) | 0.39 (+3%) | 0.51 (+3%) | 0.58 (–7%) |
| R-FCN | (×) | 0.66 (+27%) | 0.67 (+18%) | 0.69 (+11%) | 0.73 (+12%) | 0.71 (+3%) |
| Faster R-CNN | (×) | 0.68 (+14%) | 0.70 (+11%) | 0.75 (+13%) | 0.76 (+12%) | 0.81 (+10%) |

### 3.7. Seedling Size

Although it is expected that smaller seedlings are harder to detect because they are represented by a smaller area on the image, this experiment quantified the strength of this effect (Table 10).

**Table 10.** Comparison of different object detectors based on the seeding size. The metric is MAP@0.5IoU.

| Object Detector | Small | Medium | Large | All |
|:---:|:---:|:---:|:---:|:---:|
| SSD | 0.43 | 0.79 | 0.99 | 0.58 |
| SSD (aug) | 0.55 | 0.84 | 0.97 | 0.65 |
| R-FCN | 0.48 | 0.85 | 0.99 | 0.71 |
| Faster R-CNN | 0.72 | 0.91 | 1.00 | 0.81 |

All three object detectors displayed high detection rates on large seedlings. On medium seedlings, all detectors still performed reasonably well, but faster R-CNN outperformed the other detectors. Only faster R-CNN was able to reliably detect small seedlings. Figure 3 shows the precision–recall curves from faster R-CNN on small, medium, and large seedlings. The model was able to detect 100% of the large seedlings with a precision of 90%. As for medium sized seedlings, 90% of them could still be detected with a false positive rate of 20%. Unsurprisingly, small seedlings were more difficult for faster R-CNN to detect.

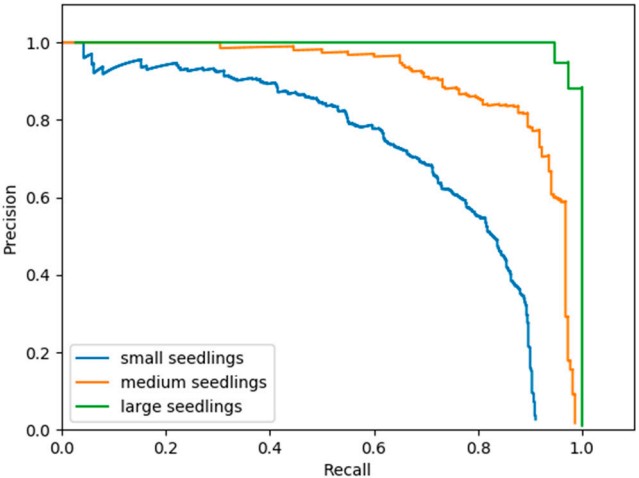

**Figure 3.** Precision–recall curves from faster R-CNN for small, medium, and large seedlings.

## 4. Discussion

The results presented in the previous section demonstrated that the proposed method could generate an accurate map of the distribution of conifer seedlings in the study area. For large and medium seedlings, the results were very accurate and thus offered a good foundation for assessing regeneration or restoration success. However, for small seedlings, the results might be not accurate enough for operational use.

The experimental evaluation indicates that MAP values of the detectors could be further increased by providing additional training images, since MAP increased with the number of samples. When employing the proposed semi-automatic annotation process of automatic prediction and manual evaluation, a new image should easily be integrated into the training data set.

Another important result is that the performance decrease due to larger ground sampling distances seemed manageable. Thus, imagery taken from higher flying altitudes covering more extended areas of seismic lines might still offer sufficient information for reliable seedling detection. For larger and medium sized seedlings, the results on images with an emulated ground sampling distance of 6.3 cm indicate that imagery taken by manned aircrafts might also provide a viable source of information.

The experiments on summer and winter images indicate that in order to train a reliable system, adding training samples representing different image contexts is beneficial. In other words, though pre-training and image augmentation help to achieve reasonable performance on limited training sets, using annotated training samples from various local and seasonal contexts should also enhance the robustness of the detection rates on new images. A similar effect occurs when applying the detectors to images with different ground sampling distances or being preprocessed in a different way. In these cases, the stability of the results relies on the detector having access to similar images during training.

Several caveats must be made regarding the generalizability of our results. The dataset we used is fairly small and homogeneous. Further tests in other regions and with other conifer species are needed to confirm general applicability. Our analysis of various spatial resolutions was based on simulated coarsened images; it would be preferable to use real images acquired at several altitudes well above the forest canopy to ascertain operational feasibility (we flew our drone inside the seismic line, which would not work for longer line segments). The reference dataset was derived from manual interpretation of the images; smaller or partially occluded seedlings might have been missed by the interpreters; a ground-based full census of seedlings with their precise location would allow a more thorough assessment of the accuracy of the detection. In the future, we envision that by combining millimetric drone imagery with SfM and CNNs, we would be able to reliably detect conifer seedlings, discard false positives on mature trees based on the photogrammetric point cloud (there were no mature trees growing in the line, but at higher flying altitudes, the photos will include the trees at the edge of the line, which may be confused with seedlings), and even estimate the height of seedlings and other attributes (e.g., species and health status). Furthermore, we speculate that by the end of next decade, when beyond visual line of sight (BVLOS) is permitted, drones carrying a graphic processing unit with a pre-trained CNN and a RTK system (real time kinetic, providing precise geolocation), could provide real-time information on stocking and spatial distribution of seedlings to field crews, who would only need to visits areas that do not meet the standards according to the drone data.

## 5. Conclusions

In this paper, we studied the use of convolutional neural networks applied to millimetric (<1 cm GSD) drone imagery to detect conifer seedlings growing on recovering linear disturbances in the boreal forest. The best architecture was a faster R-CNN detector based on a pre-trained ResNet-101 for feature map generation, which yielded a mean average precision (MAP) of 0.81. Pre-training the ResNet-101 on the COCO dataset increased the MAP by 14%. Data augmentation did not have an effect for this particular architecture. Combining images from winter and summer in the training set was beneficial for the task of object detection. Reliable detection seems possible even with a ground sampling distance of a few centimeters, especially for seedlings larger than 60 cm in crown diameter.

Our results indicate that it should be feasible to use convolutional neural networks for automated establishment surveys on sites more than five years after treatment, where the average seedling will have a sufficient size for reliable detection. However, further studies with a larger sample size are desirable to develop best practices. To conclude, the proposed method is a first step towards automated long-term forest-restoration monitoring on wide-spread legacy footprints like seismic lines.

**Author Contributions:** Conceptualization, M.F., M.S., G.M., G.C., and J.L.; methodology, M.F., M.S.; G.M., G.C., and J.L.; software, M.F.; validation, M.F.; formal analysis, M.F., investigation, M.F. and M.S.; resources, G.M., G.C. and J.L.; data curation, M.F. and M.S.; writing—original draft preparation, M.F. and G.C.; writing—review and editing, M.F., M.S., G.M., G.C. and J.L.; visualization, M.F. and J.L.; supervision, M.S., G.M., G.C. and J.L.; project administration, J.L.; funding acquisition, G.M., G.C., and J.L.

**Funding:** This research was supported by a Natural Sciences and Engineering Research Council of Canada Collaborative Research and Development Grant (CRDPJ 469943-14) in conjunction with Alberta-Pacific Forest Industries, Cenovus Energy, ConocoPhilips Canada, Canadian Natural Resources, and the Regional Industry Caribou Collaboration. Drone data collection was funded by the Office for Energy Research and Development (OERD) of Natural Resources Canada (NRCan).

**Acknowledgments:** Michael Gartrell (CFS-NoFC) acquired the drone imagery used in this study.

**Conflicts of Interest:** The authors declare no conflicts of interest.

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
