# Peer review of "Automated Detection of Conifer Seedlings in Drone Imagery Using Convolutional Neural Networks"

_remotesensing, doi:10.3390/rs11212585_

Round 1

Reviewer 1 Report

Fromme et al. provide a machine learning workflow for using drone imagery to monitor pine seedlings in disturbed areas following mining. In general most of my comments are to help improve understanding for the reader.

Comments:

Line numbers would have been useful for indicating the location for comments. 2 – sub-heading “1.2 Related work” isn’t very informative. Something like “1.2 Remote sensing of forest regeneration: Related work” would be more useful. 3 – when you say “phenological conditions (leaf-on vs. leaf-off)” do you mean of the surrounding deciduous trees that would cover conifer seedlings or for deciduous conifers (Tamarack or Larch)? The maps of Canada and Alberta in Figure 1 are really small. Could the study site be highlighted on the map of Canada and the map of Alberta removed? There would then be more room for the maps and images to be larger and easier to see. Page 5 – is flying the drone 5 m above ground level going to be realistic for a larger scale monitoring programing? Include some rationale for this height (e.g. at this point in time is it necessary to get high resolution images or can the specific drone used only fly at the height to collect useful data?). Table 2 – Please be clear as to how the four CNNs listed in Table 2 match with the three object detectors described earlier in this section. Please make this clear in Table 2/3 (e.g. make a new column) and in the section 5.1. Object Detection Architectures. I understand that this is not a primer on CNN, but the reader should not need to be well educated on CNN approaches to read this article and keep these details straight. Perhaps a workflow diagram would be helpful? The introduction is concise and short, but the methods are really dense. Is it possible to simplify them a little bit? Trim unnecessary detail so it’s a little easier to read? Table 3 – Can you spell out the paper specific acronyms in the tables and figures (e.g. COCO and MAP)? The results are similarly long compared to the discussion, which is quite succinct. Does it make sense to condense some of the tables in the results? There are some typos in the conclusions section.

Author Response

Dear Reviewer,

thanks for your valuable feedback.

Item 1 

Line numbers would have been useful for indicating the location for comments.

We neglected to add line numbers to the submitted manuscript, sorry, but are now present in the revised manuscript.

Item 2  

sub-heading “1.2 Related work” isn’t very informativeSomething like “1.2 Remote sensing of forest regeneration: Related work” would be more useful.

Changed the sub-heading accordingly to 1.2 Remote sensing of forest regeneration: Related work

Item 3

When you say “phenological conditions (leaf-on vs. leaf-off)” do you mean of the surrounding deciduous trees that would cover conifer seedlings or for deciduous conifers (Tamarack or Larch)?

None of the seedlings in the study were deciduous (tamarack or larch), so leaf-off conditions refer to the status oft he surrounding deciduous vegetation around the seedlings, not the seedling themselves.

Item 4

The maps of Canada and Alberta in Figure 1 are really small. Could the study site be highlighted on the map of Canada and the map of Alberta removed? There would then be more room for the maps and images to be larger and easier to see.

Removing the map of Alberta would not make more room, so we would prefer to leave the figure as is. In the online version, readers will be able to clck on the the ‚larger image‘ to better appreciate the details if interested.

Item 5

Page 5 – is flying the drone 5 m above ground level going to be realistic for a larger scale monitoring programing? Include some rationale for this height (e.g. at this Item in time is it necessary to get high resolution images or can the specific drone used only fly at the height to collect useful data?).

We agree with the reviewer: flying at 5 m agl is not realistic for an operational program. We actually referred to this difficulty in the discussion section, line 570-572: “it would be preferable to use real images acquired at several altitudes well above the forest canopy to ascertain operational feasibility (we flew our drone inside the seismic line, which would not work for longer line segments).” To make this clearer, we have added the following sentence in line 147 after “0.3 cm at nadir”: “We note that flying above the canopy would have resulted in imagery of a few centimeters GSD, which we initially thought may be too coarse for our application, hence our choice of flying inside the line”. We hope this provides a rationale for the chosen flight altitude.

Item 6

Table 2 – Please be clear as to how the four CNNs listed in Table 2 match with the three object detectors described earlier in this section. Please make this clear in Table 2/3 (e.g. make a new column) and in the section 5.1. Object Detection Architectures. I understand that this is not a primer on CNN, but the reader should not need to be well educated on CNN approaches to read this article and keep these details straight. Perhaps a workflow diagram would be helpful?

The three object detectors: Faster-RCNN, SSD and R-FCN describe the high-level of the model, they all use features maps (Inception v2, Resnet 51, Resnet 101, Inception-Resnet) which can be arbitrarily combined.

Item 7

Table 3 – Can you spell out the paper specific acronyms in the tables and figures (e.g. COCO and MAP)?

Fixed it in the description of Table 3

Item 8

The introduction is concise and short, but the methods are really dense. Is it possible to simplify them a little bit? Trim unnecessary detail so it’s a little easier to read?

We have streamlined a litle bit some pargraphs. However, in therms of content, we feel making further reductions could be detrimental, since we have a lot to cover.

Item 9

The results are similarly long compared to the discussion, which is quite succinct. Does it make sense to condense some of the tables in the results?

We studied different experimental settings and each setting got their own subsection to increase the readability. A further condensation oft he results might result in more confusion.

Reviewer 2 Report

page 7, 4th paragraph: Did you mean to reference Table 2 instead? "(See Table 2)", not "(See Table 1)".

Page 8, 6th paragraph: Not sure what [LAE + 15] is and where I should see it. Can you clarify?

Page 9: Figure 2 is shown in section 2.5.4 but referenced in section 2.5.5

Page 10: Figure 3 is not referenced in the body of text

Page 13: Table 4 should be moved out of the paragraph body

Page 15: Table 7 is not referenced in the body of text

Page 15: Table 8 is not referenced in the body of text

Page 15: Table 9 is not referenced in the body of text
page 16: Figure 7.1 does not exist but is referenced in the body of text

Page 17: Figure 4 is not referenced in the body of text

3.7: "it is obvious that..." I suggest rephrasing since it probably not obvious to all.

4. Discussion
"...since the MAP constantly increases with..." I suggest rephrasing to "...since the MAP increases with..."

5. Conclusions
"...indicate that it should feasible to use..." I suggest rephrasing to "...indicate that it should be feasible to use..."

Author Response

Dear Sir or Madam,

thanks for your valuable feedback.

Item 1: Page 8, 6th paragraph: Not sure what [LAE + 15] is and where I should see it. Can you clarify?

It was a missing reference, it is now [24] and referring to

[24] Liu, Wei; Anguelov, Dragomir; Erhan, Dumitru; Szegedy, Christian; Reed, Scott; Fu, Cheng-Yang; Berg, Alexander C., SSD: Single Shot MultiBox Detector, ECCV 2016 pages 21-37

Item 2: Missing references in the text

It is now fixed in the manuscript

Item 3: False labeled figures, subsections

It is now fixed in the manuscript

Item 4: Rephrasing suggestions

Adopted the phrases with the suggestions

Reviewer 3 Report

I have enjoyed reviewing this paper. It is well presented, the methodology properly explained, and results are adequately sustained by the methods and relevant for the scientific community. There are a few minor errors (such as page 6 section 2.5.1 mentions a non-existing section 2.7.2 and page 10 section 2.5.6 should be “Data Set Size” rather than “Seedling Size”) which need addressing. The paper could benefit from expanding the discussion section including potential applications such as “on-site” and “real-time” image analysis which were briefly mentioned in the manuscript but not discussed.

Author Response

Dear Reviewer,

thanks for reviewing our paper and the helpful suggestions.

Item 1: such as page 6 section 2.5.1 mentions a non-existing section 2.7.2

Response 1: It is now fixed in the manuscript

Item 2: page 10 section 2.5.6 should be “Data Set Size” rather than “Seedling Size”

Response 2: It is now fixed in the manuscript

Item 3: The paper could benefit from expanding the discussion section including potential applications such as “on-site” and “real-time” image analysis which were briefly mentioned in the manuscript but not discussed.

We have added the following text at the end of the discussion (line 582): “Furthermore, we speculate that by the end of next decade, when beyond visual line of sight (BVLOS) is permitted, drones carrying a Graphic Processing Unit with a pretrained CNN and a RTK system (real time kinetic, providing precise geolocation), could provide real-time information on stocking and spatial distribution of seedlings to field crews, who would only need to visits areas that do not meet the standards according to the drone data.”